# The Impact of Quarantine on Pain Sensation among the General Population in China during the COVID-19 Pandemic

**DOI:** 10.3390/brainsci12010079

**Published:** 2022-01-05

**Authors:** Jie Sun, Yong-Bo Zheng, Lin Liu, Shui-Qing Li, Yi-Miao Zhao, Xi-Mei Zhu, Jian-Yu Que, Ming-Zhe Li, Wei-Jian Liu, Kai Yuan, Wei Yan, Xiao-Guang Liu, Su-Hua Chang, Xuan Chen, Nan Gao, Jie Shi, Yan-Ping Bao, Lin Lu

**Affiliations:** 1Pain Medicine Center, Peking University Third Hospital, Beijing 100191, China; sunjie0615@pku.edu.cn (J.S.); shuiqingli1101@bjmu.edu.cn (S.-Q.L.); xgliu@bjmu.edu.cn (X.-G.L.); 2Peking University Sixth Hospital, Peking University Institute of Mental Health, NHC Key Laboratory of Mental Health (Peking University), National Clinical Research Center for Mental Disorders (Peking University Sixth Hospital), Chinese Academy of Medical Sciences Research Unit (No. 2018RU006), Peking University, Beijing 100191, China; yongbozheng@bjmu.edu.cn (Y.-B.Z.); linliu@bjm.edu.cn (L.L.); ximeizhu@bjmu.edu.cn (X.-M.Z.); quejianyu@bjmu.edu.cn (J.-Y.Q.); dylmzh@pku.edu.cn (M.-Z.L.); weijian191954@stu.pku.edu.cn (W.-J.L.); yuankai@pku.edu.cn (K.Y.); yanwei@bjmu.edu.cn (W.Y.); changsh@bjmu.edu.cn (S.-H.C.); 50210201017@stu.xxmu.edu.cn (X.C.); 50210105025@stu.xxmu.edu.cn (N.G.); 3Peking-Tsinghua Center for Life Sciences and PKU-IDG/McGovern Institute for Brain Research, Beijing 100191, China; 4National Institute on Drug Dependence and Beijing Key Laboratory of Drug Dependence, Peking University, Beijing 100191, China; 2011210099@stu.pku.edu.cn (Y.-M.Z.); shijie@bjmu.edu.cn (J.S.); 5School of Public Health, Peking University, Beijing 100191, China

**Keywords:** pain, COVID-19 pandemic, quarantine, mental health, China

## Abstract

During the pandemic era, quarantines might potentially have negative effects and disproportionately exacerbate health condition problems. We conducted this cross-sectional, national study to ascertain the prevalence of constant pain symptoms and how quarantines impacted the pain symptoms and identify the factors associated with constant pain to further guide reducing the prevalence of chronic pain for vulnerable people under the pandemic. The sociodemographic data, quarantine conditions, mental health situations and pain symptoms of the general population were collected. After adjusting for potential confounders, long-term quarantine (≥15 days) exposures were associated with an increased risk of constant pain complaints compared to those not under a quarantine (Odds Ratio (OR): 1.26; 95% Confidence Interval (CI): 1.03, 1.54; *p* = 0.026). Risk factors including unemployment (OR: 1.55), chronic disease history (OR: 2.38) and infection with COVID-19 (OR: 2.15), and any of mental health symptoms including depression, anxiety, insomnia and PTSD (OR: 5.44) were identified by a multivariable logistic regression. Additionally, mediation analysis revealed that the effects of the quarantine duration on pain symptoms were mediated by mental health symptoms (indirect effects: 0.075, *p* < 0.001). These results advocated that long-term quarantine measures were associated with an increased risk of experiencing pain, especially for vulnerable groups with COVID-19 infection and with mental health symptoms. The findings also suggest that reducing mental distress during the pandemic might contribute to reducing the burden of pain symptoms and prioritizing interventions for those experiencing a long-term quarantine.

## 1. Introduction

According to the WHO, a quarantine is the separation and restriction of the movement of people who have potentially been exposed to a contagious disease. Since the first outbreak in December 2019, the coronavirus disease 2019 (COVID-19) pandemic has affected 222 countries and territories, with more than 239.4 million cases and more than 4.8 million deaths reported as of 15 October 2021 [1]. Quarantines have been used as public interventions in China and worldwide [2]. Effectively implemented massive quarantines contributed to the quick containment of the epidemic, although quarantine and isolation measures might potentially have negative effects on public health, especially for those most vulnerable to chronic pain. Anticipated direct consequences of quarantines and associated traffic restrictions, including being unable to obtain access to medical care and pharmacological interventions after acute pain, uncertainty over disease status, loss of freedom, loneliness, and sleep disturbances, on occasion, create dramatic consequences.

Chronic pain affects more than 30% of people worldwide and results in significant public health and socioeconomic burdens [3]. Chronic pain was included in the top 10 leading causes of years lost to disability [4]. Depression, anxiety and insomnia are risk factors that predispose individuals to chronic pain [4] and are closely linked to deteriorated pain sensations. Increasing evidence shows that psychological distress has important roles in central pain modulating mechanisms and exacerbates persistent chronic pain [5]. Genetic correlation and shared neural abnormalities of modulatory effects of the reciprocal relationships between pain sensation and mental disorder have been recognized [6]. Although the premise that quarantines may result in a high prevalence of mental issues is underscored by a growing body of epidemiological literature [7,8,9,10] and previous research found social isolation during the pandemic made chronic pain population suffering more [11]. To our knowledge, few studies have focused on the impact of quarantines on the pain sensations among the general population. Evidence has also indicated that social interactions play an important role in the perception of pain [12,13]. Hormone stress responses were altered and adult individuals would experience more pronounced anxiety if social isolation occurred in early life [14]. Opioid overdoses were increased following the COVID-19 era [15,16], which could be a consequence of uncontrolled pain during quarantines. It is necessary for research to ascertain the prevalence of pain during quarantines and the influence of quarantines on pain sensations and identify the risk factors associated with pain to further inform interventions to mitigate pain for vulnerable groups under pandemic conditions and reduce opioid overdoses.

In our study, pain symptoms with different sites of pain were reported and compared for populations with different quarantine durations to evaluate the association of quarantine measures and pain sensation outcomes. Risk factors associated with pain among the population during COVID-19 were identified, which could serve as an evidentiary base for policy-makers to carefully weight the potential risks when developing protocols and implementing quarantines. Finally, mediation analysis was performed to illustrate that the impacts of quarantines on mental health symptoms could mediate pain sensations, which highlights the need for psychological support to relieve pain among vulnerable populations during quarantines. The primary objective of this study was to assess the prevalence of pain symptoms under the pandemic. Secondary objectives include identifying risk factors associated with pain and to explore the probable mediation effects of mental health symptoms between quarantine and pain symptoms, to provide information for the pain prevention and mental health improvement during the pandemic.

## 2. Materials and Methods

### 2.1. Study Design

The study was approved by the ethics committee of Peking University Sixth Hospital (Institute of Mental Health) (approval code: 2020-2-21-2). This cross-sectional, nationwide, web-based study was designed to investigate the psychological impacts and pain conditions related to COVID-19 among the general public in China via an online survey from 29 January to 26 April 2021 [7]. Informed consent was received online before the respondents began the questionnaire delivered through the Joybuy web portal (http://www.jd.com/ accessed on 26 April 2021). The study followed the American Association for Public Opinion Research (AAPOR) reporting guidelines [17] and the Strengthening the Reporting of Observational Studies in Epidemiology (STORBE) guidelines [18].

### 2.2. Participants

The respondents were all registered members of Joybuy, a large e-commerce and online health information service corporation with 0.44 billion active users [17,18,19]. A total of 74,588 individuals clicked on the survey link, and 38,494 respondents provided informed consent and submitted the questionnaires. A total of 4203 respondents provided repeated surveys and were further excluded. A total of 250 respondents who were younger than 18 years were also excluded because obtaining online informed consent from their parents may not be realistic. Finally, a total of 34,041 respondents were included, with a response rate of 51.6% and an effective response rate of 99.3% [20].

### 2.3. Outcomes and Covariates

The survey lasted approximately 15 min and consisted of four parts that gathered information about demographic variables, epidemic-related questions and isolation conditions, standardized mental health-related scales, and frequency of pain status. All questions in the questionnaire were introduced in a previous article [19].

The primary pain outcomes were the occurrence of the pain symptoms. Participants were asked the frequency of the pain in different locations including abdominal, headache, back, extremity and chest, suffered in the last month. Response categories were “no pain at all”, “seldom” and “frequency suffering”. The participants reported “no pain at all” or “seldom” in all above-mentioned locations were considered as a pain-free group and the participants self-reported any parts of the body with “frequency suffering pain” were considered as a group who have constant pain symptoms. Data on demographic characteristics (e.g., age, sex, educational attainment, income level, occupation, marital status, geographic location, and living area), medical comorbidities (e.g., chronic diseases and mental disorders), psychological conditions and information related to COVID-19 (e.g., infection status of participants; whether the participants were engaged in frontline work related to COVID-19 including medical care, scientific research, disease control and management, and supply support; status of work or school resumption; experience with public health interventions including quarantines, traffic restrictions and community confinement; and whether the participants were unemployed due to COVID-19 pandemic) were collected via questionnaires. The questionnaire were set based on legally requirements of quarantine policy [21]. We divided quarantines into no quarantines, short-term quarantines (≤7 days), mid-term quarantines (8–14 days) and long-term quarantines (≥15 days) according to the duration time. Any mental symptom of depression, anxiety, insomnia and PTSD was considered mental health symptoms. We dichotomized mental health into either a mentally healthy group (0) or any with mental health symptoms group (1). Chinese versions of the Patient Health Questionnaire-9 (PHQ-9) [22], Generalized Anxiety Disorder-7 (GAD-7) [23], Insomnia Severity Index(ISI) [24] and Post-traumatic Stress Disorder Checklist for DSM-5 (PCL-5) [25], which measure symptoms of depression, anxiety, insomnia, and acute stress, respectively, were used to represent psychological status. PHQ-9, GAD-7 and ISI inquires about depression, anxiety and insomnia symptoms in the past 2 weeks while PCL-5 inquires about PTSD symptoms in the recent month. The cutoff scores were 5, 5, 8, and 33 for detecting depression, anxiety, insomnia and posttraumatic stress disorder (PTSD) symptoms, respectively, according to a previous study [7,19,20]. Scores of participants greater than cutoff threshold indicate potential mental health symptoms. 

### 2.4. Statistical Analysis

Analyses were conducted using the IBM SPSS Statistics 26.0 (IBM Corp., Armonk, NY, USA) and Mplus 8 software (Muthen & Muthen Corp., Los Angeles, CA, USA). Descriptive statistics were used to analyze demographic characteristics and pandemic-related information. The prevalence of pain symptoms was reported as percentages of cases in different populations among all and quarantined populations. χ^2^ tests were used to compare the prevalence of different pain symptoms in stratified populations. The Mann–Whitney *U* test was used to compare continuous variables not normally distributed. Multivariable logistic regression analysis was performed to calculate the adjusted odds ratios (ORs) and 95% confidence intervals (CIs) of the risk of pain symptoms among all and long-term quarantined people after adjusting for potential confounders, including demographic characteristics (age, gender, economic status, marriage status, education attainment, living areas, and income), medical conditions (chronic disease, COVID-19 infection status, and mental health issues), and epidemic-related factors (experience with public health interventions).

The hypothesis of the mediating effect of mental health symptoms between quarantine and pain symptoms was investigated using a 4-step analysis with a bootstrap approach [26]. The method involved testing a direct path between quarantine status and pain and then estimating how much the association is mediated by mental health symptoms. Following multiple categorical variate mediation analysis, bootstrapping methods were used to verify the indirect effect and to produce bias-corrected confidence intervals, which were based on the confidence intervals after 5000 bootstrapping resamples. Statistical significance was set at *p* < 0.05, and all tests were two-tailed.

## 3. Results

### 3.1. Demographic Characteristics

A total of 34,041 participants from 34 provinces in China completed this cross-sectional, nationwide study. Of the total sample, 15,732 participants (46.2%) were male, and 20,957 (60.9%) participants were 18–39 years. Furthermore, 26,957 participants (79.2%) had a university degree or higher, and 26,392 (77.5%) were married. Of the total number of respondents, 20,727 (60.9%) were aged 18 to 39 years, and 26,492 (79.2%) participants lived in urban areas. This survey included data from 104 individuals (0.3%) with confirmed or suspected cases of COVID-19. Most of the participants (33,873 participants (99.5%)) did not have a history of mental illness. A total of 11,947 (35.1%) participants had experience with community containment. Additional demographic and epidemic-related characteristics are presented in Table 1.

### 3.2. Pain Status under Different Quarantine Durations

The prevalence of constant pain symptoms among the total sample was 6.9%: 1.4% for abdominal pain, 2.1% for headache pain, 3.6% for extremity pain, 1.1% for chest pain, and 2.8% for back pain. A total of 9.4% of participants (120/1282) felt constant pain after long-term quarantines while 6.7% of participants (2014/30160) felt constant pain and did not experience a quarantine (Table 2). Interestingly, we found that the ORs increased with the quarantine duration for all types of pain symptoms. The risk of constant pain symptoms increased with the cumulative quarantine duration (Table 3). Those who underwent mid-term quarantines and long-term quarantines had a significantly higher risk of headache pain and chest pain, as presented in Table 3. Participants who experienced long-term quarantine measures reported higher prevalence rates of constant extremity pain and chest pain. Participants who experienced mid-term quarantine measures reported a higher risk of constant chest pain and headache.

### 3.3. Risk Factors Associated with Constant Pain Symptoms

The multivariable logistic regression found that after adjusting for potential confounders, those aged 40–59 (OR: 1.33; 95% CI: 1.21, 1.46; *p* < 0.001); those who underwent long-term quarantines (OR: 1.26; 95% CI: 1.03, 1.54; *p* = 0.026); those who had a college degree or higher (OR: 1.21; 95% CI: 1.07, 1.36; *p* = 0.002); those having mental health issues including depression, anxiety, insomnia and PTSD (OR: 5.44; 95% CI: 4.91, 6.02; *p* < 0.001); those who were infected or suspected of being infected by COVID-19 (OR: 2.15; 95% CI: 1.28, 3.62; *p* = 0.004); those having a history of chronic disease (OR: 2.38; 95% CI: 2.11, 2.67; *p* < 0.001); frontline workers (OR: 1.27; 95% CI: 1.14, 1.42; *p* < 0.001); and those who lost their jobs due to the pandemic (OR: 1.55; 95% CI: 1.36, 1.78; *p* < 0.001) had significantly higher rates of constant pain symptoms. Compared to females, males had lower risk (OR: 0.59; 95% CI: 0.54, 0.64; *p* < 0.001) of constant pain symptoms. Living in rural areas resulted in a lower risk of constant pain symptoms (OR: 0.85; 95% CI: 0.76, 0.96; *p* < 0.001, Table 4).

In order to identify the risk factors for constant pain symptoms after long-term quarantines, we conducted a multivariable logistic regression of participants who experienced quarantines that exceeded 14 days. Multivariable logistic regression analysis revealed that after adjusting for potential confounders, those 40–59 years old (OR: 1.64; 95% CI: 1.03, 2.6; *p* = 0.037), those having a history of chronic diseases (OR: 2.02; 95% CI: 1.17, 3.5; *p* = 0.012), those who lost jobs due to COVID-19 (OR: 1.82, 95% CI: 1.13, 2.95, *p* = 0.015), those who were infected or suspected of being infected by COVID-19 (OR: 3.79, 95% CI: 1.15, 12.49, *p* = 0.029), and those with any mental health symptoms (OR: 6.49, 95% CI: 3.83, 10.97, *p* < 0.001) had a significantly higher risk of pain complaints. Compared with those with household income below 5000 RMB, participants with higher household income reported a lower risk of pain symptoms (OR: 0.47, 95% CI: 0.23, 0.95, *p* = 0.036) (Table 4).

### 3.4. Mental Health Symptoms Could Mediate the Impact of Quarantines on Pain Symptoms

Furthermore, we explored the mediating function of any mental health symptom between quarantine exposure and pain symptoms. In the mediation analysis with mental health symptoms as the mediator, the total (β = 0.132, *p* = 0.00) and indirect (β = 0.075, *p* = 0.00) effects of quarantine duration on pain sensation were significant, and the direct effect of quarantine duration on pain symptoms was also significant (β = 0.056, *p* = 0.04). The results suggested that mental health symptoms may partially account for the relationship between quarantine duration and pain symptoms (Figure 1).

## 4. Discussion

Quarantines were first adopted to combat leprosy in Venice, Italy in 1127 and are widely imposed to urgently control pandemics [8]. Implementation of quarantine procedure has been employed to fight against plague [27,28], such as tuberculosis, SARS, etc., although quarantine is not a panacea, it has its limits. This is one of the largest nationwide studies conducted to investigate the role of mental health symptoms as a mediator of the relationship between quarantines and constant pain in general Chinese populations. Those who suffered long-term quarantines experienced a greater risk than those who were not quarantined, indicating long-term quarantine had an adverse effect on pain conditions. Increasing pain risk is associated with long-term quarantines, notably for vulnerable groups who are infected or suspected of being infected by COVID-19, those with chronic disease history, those who were unemployed due to the pandemic, and those experiencing a mental health burden. In addition, the mediation analysis showed that mental health symptoms play a significant role in the association of quarantine exposure and constant pain symptoms. These exploratory findings may imply that certain demographics and mental health symptoms convey higher risk under long-term quarantines. Curated and targeted interventions could protect the general population from mental health symptoms to reduce the chronic pain burden.

### 4.1. The Prevalence of Pain during the Pandemic

Well-prepared regular healthy food instead of takeout food and decreasing sedentary working hours might counteract the negative effects of quarantines, which might affect people’s digestion and back pain. Anxiety, depression and other mental issues could be behind a substantial portion of chest pain while heart disease is only one of several causes of chest pain and is the least common [29]. Chest pain and tension headache pain stem from unconsciously constant tightening of certain muscles due to mental health symptoms. Thus, it was not surprising to see that quarantines impact these two types of pain.

### 4.2. Risk Factors for Chronic Pain during the Pandemic Era

Mental health issues, including depression, anxiety, PTSD and insomnia, were the strongest predictors for the subsequent development of constant pain during the pandemic. Other subsequent exploratory assessments of factors that might increase constant pain have overlapping risk factors of chronic pain [30].

Gate-control theory [31] might help to explain the accumulative response effect of quarantine duration on the risk of constant pain symptoms. A long-term quarantine duration might facilitate the flow of nerve impulses to the brain by opening the door that diminishes the transmission of pain information while short/mid-term quarantines might not trigger the conversion of pain sensation from a normal to a constant pain state. In people who experienced long-term quarantine, unlike the studies before the pandemic [30], we found people who had mental health issues had 6.49 more relative risk of pain than people who were mentally healthy. 

We identified several characteristics associated with constant pain symptoms: chronic disease history, being infected by COVID-19, unemployment due to the pandemic and with mental health symptoms during the pandemic. During quarantine, patients with chronic disease could not have access to the medical resources as usual. Adverse events of drugs or uncontrolled physical status might worsen the health condition of chronic disease patients under quarantine [32,33]. Without the prompt and high quality medical care [34,35,36], chronic disease might increase the risk of chronic pain. 

### 4.3. Characteristics Impacting Constant Pain Symptoms

#### 4.3.1. Mental Health Symptoms

We further found that mental health symptoms further mediated the indirect effect of long-term quarantine duration on pain symptoms (Figure 1). Quarantines have considerable psychological impacts, although they are an effective measure to prevent the spread of pandemic [7,8,19,37,38], which was also shown as an effect in our mediation analysis model. We found that mental health symptoms were the largest risk factors for constant pain in the long-term quarantine population, indicating that pain could be a consequence of those mental health symptoms (effect b in Figure 1). These basic findings are consistent with research showing that psychological factors that are frequently comorbid with chronic pain also predispose patients to the development of chronic pain [4,6] and may in turn exacerbate and worsen pain symptoms. No social obligations and the lack of communication during quarantines did not result in more sleep; conversely, most people actually experienced worse quality sleep due to their damaged sleep schedules, stress and anxiety [39]. Although it is widely known that pain can cause mental health symptoms and sleep disturbances, we also need to realize that the interaction is bidirectional. Sleep disturbance could exacerbate pain via neurophysiological processes that modulate pain signaling at supraspinal and spinal levels [6]. 

Social isolation disrupts the fight-or-fight response by leading to a decrease in white blood cells and increasing inflammation [40,41]. Importantly, the mediation analysis in our research suggested that pain rendered physical symptoms the denouement of a dynamic interaction between biological, psychological, and social factors, which reciprocally influenced each other, consistent with the biopsychosocial model of pain [42]. A deeper understanding of the effects that quarantines and a person’s mental health symptoms can have on the influence of their somatic complaints is necessary.

#### 4.3.2. COVID-19 Infection

Chronic malaise, diffuse myalgia and nonrestorative sleep were reported by COVID-19 survivors [43], as seen with influenza and noted in the H1N1 pandemic and SARS epidemic [44,45]. We found those who were diagnosed or suspected of infection had a higher (OR: 3.79) risk of constant pain. Health care systems and providers should be prepared to recognize and meet the ongoing needs of postpandemic chronic pain.

#### 4.3.3. Financial Problems

Interlinked with environmental factors, upbringing, nutrition, socioeconomic status, and income were specifically associated with chronic pain [46,47]. Consistent with that, our research found that higher household incomes were related to a lower risk of constant pain. Moreover, we found that those who lost jobs due to the pandemic reported a higher risk of constant pain. As a consequence of quarantines, financial loss or unemployment created serious socioeconomic distress [48]. The link between socioeconomic status and pain in a long-term quarantine population provides insights into the socioeconomic influences on the pain biopsychosocial model and further substantiates that the chronic pain after long-term quarantine is not only rooted in biology but also intimately embedded in society.

### 4.4. What Can Be Done to Mitigate the Constant Pain Consequence of Quarantines?

#### 4.4.1. Formulate Precise Quarantine Policy

Long-term comprehensive quarantines are related to constant pain as the factors for chronic pain risk could have more of an effect the longer they were experienced. Restricting the previous location and appropriate length of quarantines to what is scientifically legitimate given the known duration of latent periods could reduce the burden of chronic pain. Extension of the quarantine might exacerbate the sense of frustration or demoralization [49], which might aggravate the sense of pain. Adhering to the precious quarantine strategy and adequate social support during isolation are important.

#### 4.4.2. Provide Psychological and Social Support during Quarantines to Promote Mental Health

Since the strongest risk factor for constant pain in long-term quarantines was mental health symptoms, countermeasures should be considered to mitigate mental health symptoms during long-term quarantines. Longer quarantine durations were correlated with poorer mental health [50,51]. People who experienced long-term quarantines were exposed to lengthy and jumbled media coverage; and worries about being infected, unmet medical needs, job concerns, and economic issues are pervasive [52]. Catastrophizing was shown to mediate the relationship between negative events and pain symptoms [53]. Catastrophic appraisals of any physical symptoms experienced during quarantines might increase the risk of chronic pain in the fear-avoidance model. Giving mental support and having a robust social network would provide a “pain-buffering” benefit.

#### 4.4.3. Offer Financial Aid

An analysis of post-COVID-19 prospects projects that global unemployment will reach 205 million in 2022. Bolstering the economy could relieve people’s burdens and create more job opportunities under COVID-19, which might diminish the prevalence of chronic pain by creating a leverage effect to benefit individuals’ income, social welfare and health care. People who are quarantined and have lost their jobs might require additional levels of support. 

### 4.5. Future Directions

Mediation analysis suggests that interdisciplinary medical cooperation, including psychosis physicians and pain physicians, is needed for chronic pain patients enduring long-term quarantines. Given the global scale of the quarantines implemented to combat this pandemic, it is apparent that the health care needs for patients with pain syndromes or mental distress sequelae of COVID-19 will continue to increase. It is important to harness the existing pain center infrastructure, develop scalable health models and integrate the models across multidisciplinary team to achieve improved mental health and alleviate the physical pain of people postpandemic in the long term.

### 4.6. Strengths and Limitations

To our knowledge, this is the first study using a large, nationwide population-based survey to investigate the longstanding impact of quarantine on pain symptoms and the associated risk factors that may contribute to constant pain. Another important finding was the mental health symptoms played a critical role in mediating the quarantine duration effects on constant pain symptoms. Our data complement the model that quarantine has significant pain symptoms and mental health symptoms consequences. 

Our current study implemented a cross-sectional design and lacked data before quarantines, which might make the model incapable of determining whether quarantine measures have long-term consequences for chronic pain. Longitudinal studies and before-and-after designs are warranted to clarify whether quarantines impact pain sensations and whether these outcomes will have long-lasting effects after quarantines. Participants with a chronic pain history before the pandemic should be excluded from the analysis in the longitudinal studies. In addition, primary pain outcomes were subjective and self-reported by participants rather than through the clinical diagnosis or objective examination, which might be particularly susceptible to confounding factors. Finally, although the response rate and completion rate were 51.6% and 99.3%, respectively, we acknowledge that a considerable risk of bias might have substantially influenced the results. The results should be interpreted in the context of the potential limitations, including concerns about bias and generalizability to other study designs or populations.

## 5. Conclusions

Although the public benefits of quarantines are well-established, psychosocial consequences related to quarantines may place individuals at a heightened likelihood of chronic pain. Our study using a large, nationwide population-based survey provide novel evidence that long-term quarantine had a negative influence on pain symptoms. It is obvious that the relationship between chronic pain, confinement-related mental disorders, and quarantine effects could be dramatically interplayed with a further impairment of their clinical conditions and quality of life in general. The toll of long-term quarantines extends beyond psychosocial stressors that include prolonged periods of isolation, fear of illness, and financial strain with important constant somatic pain symptoms.

## Figures and Tables

**Figure 1 brainsci-12-00079-f001:**
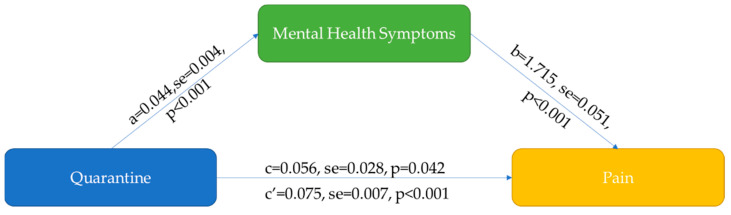
Mental health symptoms mediated the relationship between quarantine and pain. For the mediation model, Quarantine is an independent variable, Pain is an outcome variable, Mental health symptoms is a mediation variable. Path “a” is the effect of Quarantine on Mental health symptoms, path “b” is the effect of Mental health symptoms on Pain, path “c” is the effect of Quarantine on Pain (direct effect), “c’” is the indirect effect of Quarantine on Pain.

**Table 1 brainsci-12-00079-t001:** Demographic characteristics of respondents by quarantine condition.

Characteristics	Total, No. (%)	Quarantine Condition, No. (%)			
		No	≤7 Days	8–14 Days	≥15 Days
Overall	34,041 (100.0)	30,160 (88.6)	709 (2.1)	1890 (5.6)	1282 (3.8)
Gender					
Male	15,732 (46.2)	13,781 (45.7)	355 (50.1)	972 (51.4)	624 (48.7)
Female	18,309 (53.8)	16,379 (54.3)	354 (49.9)	928 (48.6)	658 (51.3)
Age					
18–39	20,727 (60.9)	17,807 (59.0)	526 (74.2)	1441 (76.2)	953 (74.3)
40–59	12,713 (37.3)	11,788 (39.1)	170 (24.0)	440 (23.3)	315 (24.6)
>60	601 (1.8)	565 (1.9)	13 (1.8)	9 (0.5)	14 (1.1)
Marriage status					
Married	26,392 (77.5)	23,741 (78.7)	489 (69.0)	1289 (68.2)	873 (68.1)
Unmarried ^a^	7649 (22.5)	6419 (21.3)	220 (31.0)	601 (31.8)	409 (31.9)
Education attainment					
<College/ undergraduate	7084 (20.8)	6213 (20.6)	154 (21.7)	407 (21.5)	310 (24.2)
≥College/ undergraduate	26,957 (79.2)	23,947 (79.4)	555 (78.3)	1483 (78.5)	972 (75.8)
Living areas					
Urban	26,942 (79.1)	23,978 (79.5)	552 (77.9)	1445 (76.5)	967 (75.4)
Rural	7099 (20.9)	6182 (20.5)	157 (22.1)	445 (23.5)	315 (24.6)
Household income, RMB ^b^					
<5000	8438 (24.8)	7302 (24.2)	222 (31.3)	519 (27.5)	395 (30.8)
5000–12,000	15,961 (46.9)	14,150 (46.9)	320 (45.1)	896 (47.4)	595 (46.4)
>12,000	9642 (28.3)	8708 (28.9)	167 (23.6)	475 (25.1)	292 (22.8)
History of chronic diseases					
No or unknown	30,938 (90.9)	27,382 (90.8)	662 (93.4)	1744 (92.3)	1150 (89.7)
Yes	3103 (9.1)	2778 (9.2)	47 (6.6)	146 (7.7)	132 (10.3)
Unemployment due to COVID-19					
No or unknown	27,898 (92.5)	619 (87.3)	1648 (87.2)	1088 (84.9)	31,253 (91.8)
Yes	2262 (7.5)	90 (12.7)	242 (12.8)	194 (15.1)	2788 (8.2)
Infection status of COVID-19					
Uninfected	33,937 (99.7)	30,107 (99.8)	696 (98.2)	1868 (98.8)	1266 (98.8)
Suspected or diagnosed	104 (0.3)	53 (0.2)	13 (1.9)	22 (1.1)	16 (1.2)
Participation of frontline work related to the outbreak ^c^					
No	28,261 (83.0)	25,251 (83.7)	536 (75.6)	1454 (76.9)	1020 (79.6)
Yes	5780 (17.0)	4909 (16.3)	173 (24.4)	436 (23.1)	262 (20.4)
Mental health symptoms					
No	20,093 (59.0)	18,195 (60.3)	316 (44.6)	923 (48.8)	659 (51.4)
Yes	13,948 (41.0)	11,965 (39.7)	393 (55.4)	967 (51.2)	623 (48.6)

^a^ The unmarried category included separated, divorced, and widowed. ^b^ As of 24 June 2021, 1 RMB = USD $0.15. ^c^ Frontline work related to the outbreak included healthcare, work associated with epidemic prevention and control, cold chain import, public service.

**Table 2 brainsci-12-00079-t002:** Pain conditions characteristics under different quarantine duration.

Characteristics	Frequency	Total, No. (%)	Quarantine Condition, No. (%)			
			No	≤7 Days	8–14 Days	≥15 Days
Stomachache	Never/seldom	33,564 (98.6)	29,765 (98.7)	697 (98.3)	1850 (97.9)	1252 (97.7)
	Regularly	477 (1.4)	395 (1.3)	12 (1.7)	40 (2.1)	30 (2.3)
Headache	Never/seldom	33,327 (97.9)	29,572 (98.1)	688 (97.0)	1828 (96.7)	1239 (96.6)
	Regularly	714 (2.1)	588 (1.9)	21 (3.0)	62 (3.3)	43 (3.4)
Backache	Never/seldom	33,077 (97.2)	29,331 (97.3)	684 (96.5)	1827 (96.7)	1235 (96.3)
	Regularly	964 (2.8)	829 (2.7)	25 (3.5)	63 (3.3)	47 (3.7)
Extremity pain	Never/seldom	32,805 (96.4)	29,106 (96.5)	684 (96.5)	1809 (95.7)	1206 (94.1)
	Regularly	1236 (3.6)	1054 (3.5)	25 (3.5)	81 (4.3)	76 (5.9)
Chest pain	Never/seldom	33,673 (98.9)	29,878 (99.1)	697 (98.3)	1847 (97.7)	1251 (97.6)
	Regularly	368 (1.1)	282 (0.9)	12 (1.7)	43 (2.3)	31 (2.4)
Pain sites	0	31,698 (93.1)	28,146 (93.3)	657 (92.7)	1733 (91.7)	1162 (90.6)
	1	1514 (4.4)	1321 (4.4)	31 (4.4)	88 (4.7)	74 (5.8)
	>1	829 (2.5)	693 (2.3)	21 (2.9)	69 (3.7)	46 (3.6)

**Table 3 brainsci-12-00079-t003:** The impact of quarantine duration on different constant pain site.

		Unadjusted OR (95% CI)	*p* Value	Multivariable Adjusted OR (95% CI) ^a^	*p* Value
Pain condition	No	1 [Reference]		1 [Reference]	
	≤7 days	1.11 (0.83, 1.47)	0.489	0.91 (0.68, 1.22)	0.519
	8–14 days	1.27 (1.07, 1.5)	0.006	1.11 (0.93, 1.33)	0.233
	≥15 days	1.44 (1.19, 1.75)	<0.001	1.26 (1.03, 1.54)	0.026
Stomachache	No	1 [Reference]		1 [Reference]	
	≤7 days	1.3 (0.73, 2.32)	0.378	0.92 (0.51, 1.66)	0.776
	8–14 days	1.63 (1.17, 2.26)	0.004	1.24 (0.88, 1.74)	0.215
	≥15 days	1.81 (1.24, 2.63)	0.002	1.35 (0.92, 1.99)	0.126
Headache	No	1 [Reference]		1 [Reference]	
	≤7 days	1.54 (0.99, 2.39)	0.057	1.10 (0.7, 1.73)	0.685
	8–14 days	1.71 (1.31, 2.23)	<0.001	1.36 (1.03, 1.79)	0.029
	≥15 days	1.75 (1.27, 2.39)	0.001	1.37 (0.99, 1.9)	0.058
Backache	No	1 [Reference]		1 [Reference]	
	≤7 days	1.29 (0.86, 1.94)	0.213	0.99 (0.66, 1.5)	0.966
	8–14 days	1.22 (0.94, 1.58)	0.135	1.00 (0.77, 1.31)	0.991
	≥15 days	1.35 (1, 1.82)	0.051	1.10 (0.81, 1.5)	0.543
Extremity pain	No	1 [Reference]		1 [Reference]	
	≤7 days	1.01 (0.67, 1.51)	0.964	0.90 (0.59, 1.36)	0.616
	8–14 days	1.24 (0.98, 1.56)	0.072	1.19 (0.93, 1.51)	0.164
	≥15 days	1.74 (1.37, 2.21)	<0.001	1.63 (1.27, 2.1)	<0.001
Chest pain	No	1 [Reference]		1 [Reference]	
	≤7 days	1.82 (1.02, 3.27)	0.043	1.07 (0.59, 1.95)	0.822
	8–14 days	2.47 (1.78, 3.41)	<0.001	1.68 (1.2, 2.36)	0.002
	≥15 days	2.63 (1.8, 3.82)	<0.001	1.8 (1.22, 2.65)	0.003

Abbreviations: OR, odds ratio; CI, confidential interval. ^a^ Adjusted for gender, age, marriage, education attainment, living areas, comorbidity of chronic diseases, household income, unemployment due to the pandemic, infection status of COVID-19, quarantine duration, mental health symptoms and participation of frontline work related to the outbreak.

**Table 4 brainsci-12-00079-t004:** Risk factors for pain condition in participants.

Variable	No. of Cases with Pain/No. of Total Population (%)	Adjusted OR (95% CI) ^a^	*p* Value	No. of Cases with Pain/No. of Long-Term Quarantine (%)	Adjusted OR (95% CI) ^b^	*p* Value
Gender						
Male	1471/18,309 (8.0)	1 [Reference]	NA	64/658 (9.7)	1 [Reference]	NA
Female	872/15,732 (5.5)	0.59 (0.54, 0.64)	<0.001	56/624 (9)	0.79 (0.52, 1.19)	0.257
Age						
18–39	1310/20,727 (6.3)	1 [Reference]	NA	78/953 (8.2)	1 [Reference]	NA
40–59	995/12,713 (7.8)	1.33 (1.21, 1.46)	<0.001	41/315 (13)	1.64 (1.03, 2.6)	0.037
>60	38/601 (6.3)	0.93 (0.65, 1.32)	0.680	1/14 (7.1)	0.60 (0.07, 5.09)	0.636
Marriage status						
Married	1761/26,392 (6.7)	1 [Reference]	NA	85/873 (9.7)	1 [Reference]	NA
Unmarried	582/7649 (7.6)	1.11 (1.00, 1.24)	0.046	35/409 (8.6)	0.75 (0.48, 1.18)	0.218
Education attainment						
Less than college	433/7084 (6.1)	1 [Reference]	NA	31/310 (10)	1 [Reference]	NA
College degree or higher	1910/26,957 (7.1)	1.21 (1.07, 1.36)	0.002	89/972 (9.2)	1.21 (0.74, 1.98)	0.44
Living areas						
Urban	1930/26,942 (7.2)	1 [Reference]	NA	87/967 (9)	1 [Reference]	NA
Rural	413/7099 (5.8)	0.85 (0.76, 0.96)	0.007	33/315 (10.5)	1.14 (0.71, 1.82)	0.591
Household income						
<5000	586/8438 (6.9)	1 [Reference]	NA	41/395 (10.4)	1 [Reference]	NA
5000–12,000	1070/15,961 (6.7)	1.02 (0.91, 1.13)	0.800	66/595 (11.1)	1.26 (0.81, 1.98)	0.307
>12,000	687/9642 (7.1)	1.13 (1.00, 1.29)	0.056	13/292 (4.5)	0.47 (0.23, 0.95)	0.036
History of chronic diseases						
No or unknown	1868/30,938 (6)	1 [Reference]	NA	96/1150 (8.3)	1 [Reference]	NA
Yes	475/3103 (15.3)	2.38 (2.11, 2.67)	<0.001	24/132 (18.2)	2.02 (1.17, 3.5)	0.012
Unemployment due to COVID-19						
No or unknown	2035/31,253 (6.5)	1 [Reference]	NA	87/1088 (8)	1 [Reference]	NA
Yes	308/2788 (11.0)	1.55 (1.36, 1.78)	<0.001	33/194 (17)	1.82 (1.13, 2.95)	0.015
Infection status of COVID-19						
Uninfected	2323/33,937 (6.8)	1 [Reference]	NA	114/1266 (9)	1 [Reference]	NA
Diagnosed or suspected	20/104 (19.2)	2.15 (1.28, 3.62)	0.004	6/16 (37.5)	3.79 (1.15, 12.49)	0.029
Participation of frontline work related to the outbreak						
No	1863/28,261 (6.6)	1 [Reference]	NA	88/1020 (8.6)	1 [Reference]	NA
Yes	480/5780 (8.3)	1.27 (1.14, 1.42)	<0.001	32/262 (12.2)	1.28 (0.79, 2.07)	0.311
Mental health symptoms						
No	525/20,093 (2.6)	1 [Reference]	NA	18/659 (2.7)	1 [Reference]	NA
Yes	1818/13,948 (13.0)	5.44 (4.91, 6.02)	<0.001	102/623 (16.4)	6.49 (3.83, 10.97)	<0.001
Quarantine duration						
No	2014/30,160 (6.7)	1 [Reference]	NA			
≤7 days	52/709 (7.3)	0.91 (0.68, 1.22)	0.519			
8–14 days	157/1890 (8.3)	1.11 (0.93, 1.33)	0.233			
≥15 days	120/1282 (9.4)	1.26 (1.03, 1.54)	0.026			

Abbreviations: OR, odds ratio; CI, confidential interval; NA, not available. ^a^ Adjusted for gender, age, marriage, education attainment, living areas, comorbidity of chronic diseases, household income, unemployment due to the pandemic, infection status of COVID-19, quarantine duration, mental health symptoms and participation of frontline work related to the outbreak. ^b^ Adjusted for gender, age, marriage, education attainment, living areas, comorbidity of chronic diseases, household income, unemployment due to the pandemic, infection status of COVID-19, mental health symptoms and participation of frontline work related to the outbreak.

## Data Availability

The data for this article will be shared on reasonable request to the corresponding author.

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
