# Peer review of "The Impact of Quarantine on Pain Sensation among the General Population in China during the COVID-19 Pandemic"

_brainsci, 2022, doi:10.3390/brainsci12010079_

Round 1

Reviewer 1 Report

In this manuscript, the authors investigated the effect of quarantine on pain symptoms in the general population in China during the covid-19 pandemic. They also tested the mediating role of mental health symptoms in this effect. The topic is timely and the story rather straightforward. The manuscript, however, did not describe the methodologies clearly, especially regarding the statistical analysis part.

1), the first main issue is it is unclear how the authors used the primary outcome for the logistic regression analysis. They mentioned nothing about how they combined the five different kinds of pains and whether the number of pain sites or the duration of the pain condition was used for the multivariate logistic regression analysis.

2), the variable of mental healthy symptoms also needs clarification. So the score of this variable ranges from 0 to 4? Each scale used here should also have a proper citation and a brief description especially regarding on how long a time scale does each inventory measures their respect symptoms, current, past one week, two weeks, one month?

3), as an online survey, did the authors include any screening items? what did they ensure the quality of the survey or that all subjects responded in a reliable and authentic way?

4), Abstract: in line 33, is it > 14 days or > 15 days? the effect size should also be reported here. The definition of mental health symptoms here is unclear and should be clarified. The effect sizes or ORs of the risk factors as well as the effect size of the mediation effect should also be reported here.

Reviewer 2 Report

Dear authors thank you for allowing me to review this work. The article is generally well written and well-conducted considering the cross-sectional study design. However, I think there are a few things that can be improved.

  • In the introduction, it would be great if the authors report and cite the effects of quarantine in pain populations. There are many studies out here of the impact of COVID-19 on chronic pain patients including evaluations of quarantine measures.
  • In the methods the description of quarantine measurements is not well explained. The author stated that they divide them into no quarantines, short-term quarantines (<7 days), mid-term quarantines (8-14 days) and long-term quarantines (>15 days) according to the duration time. However, quarantine policies have different levels of restrictions, and this point is not well established.
  • Also, the authors did not report any inclusion or exclusion criteria for exposed and unexposed subjects despite the cross-sectional study.
  • Please Indicate if evaluators of subjective components of the study were masked to other aspects of the status of the participants
  • Please also explain any participant exclusions from the analysis
  • if applicable, explain how missing data were handled in the analysis

Reviewer 3 Report

  • Lines 70-79: Among these points you report, which of these are the primary and secondary objectives? Please state.
  • Lines 37-39: "Quarantines were first adopted to combat leprosy in Venice, Italy in 1127 and... exposed to a contagious disease" it is better that this historical vignette is moved to the beginning of the discussion, and also improved. Look at these 2 refs.  -- The contribution of Carlo Giacomini (1840-1898): the limbus Giacomini and beyond. Neurosurgery. 2013 Mar;72(3):475-81; discussion 481-2. doi: 10.1227/NEU.0b013e31827fcda3.   --  The concept of quarantine in history: from plague to SARS. J Infect. 2004 Nov;49(4):257-61. doi: 10.1016/j.jinf.2004.03.002. 
  • Lines 105-106: "The primary pain outcomes were the frequency of the pain, including abdominal pain, headache pain, back pain, extremity pain and chest pain, respectively, that occurred last month" What  do authors mean?
  • Table 1 show that the presence of "mental health symptoms" in patients which stay at home for more than 14 days is similar. how do authors explain this results?
  • But in table 4, it seems that quarantine duration more than 15 days is statistically significant on pain and related neurological pathologies.
  • Lines 248-256. The importance of chronic disease during pandemic  that are expanded but less discussed, authors have to speak about. Improve this point, look: doi: 10.1007/s40263-013-0135-1  --  doi: 10.1302/0301-620X.102B9.BJJ-2020-1147.R1  --  doi: 10.1016/j.bbi.2020.11.020  -- doi: 10.1016/j.bbi.2020.11.034   -- doi: 10.1016/j.amjopharm.2007.10.002
  • Lines 313-314: Based on which data, do authors suggest to "minimize the quarantine time" (paragraph 4.4.1) ? explain.
  • Explain better figure 1. Add more details in the figure legend.
  • Lines 354-356: "Strengths and Limitations -- To our knowledge, this is the first study using a large, nationwide population-based... associated risk factors that may contribute to constant pain." are authors sure about that?

Round 2

Reviewer 1 Report

Thank the authors for addressing my concerns.

Reviewer 3 Report

Good job.